# Multi-Analytical Insight into the Non-Volatile Phytochemical Composition of *Coleus aromaticus* (Roxb.) Benth.

**DOI:** 10.3390/metabo16010015

**Published:** 2025-12-23

**Authors:** Chiara Toniolo, Martina Bortolami, Adriano Patriarca, Daniela De Vita, Fabio Sciubba, Luca Santi

**Affiliations:** 1Department of Environmental Biology, Sapienza University of Rome, Piazzale Aldo Moro 5, 00185 Rome, Italy; martina.bortolami@uniroma1.it (M.B.); daniela.devita@uniroma1.it (D.D.V.); fabio.sciubba@uniroma1.it (F.S.); l.santi@uniroma1.it (L.S.); 2Department of Chemistry, University of Rome Sapienza, Piazzale Aldo Moro 5, 00185 Rome, Italy; adriano.patriarca@uniroma1.it; 3NMR-Based Metabolomics Laboratory (NMLab), Sapienza University of Rome, Piazzale Aldo Moro 5, 00185 Rome, Italy; 4Interdepartmental Center of Applied Sciences for the Protection of the Environment and Cultural Heritage (CIABC), Sapienza University of Rome, Piazzale Aldo Moro 5, 00185 Rome, Italy

**Keywords:** *Coleus aromaticus*, non-volatile metabolites, amino acids, polyphenols, organic acids, chromatography, NMR

## Abstract

**Background/Objectives:** *Coleus aromaticus* (Lamiaceae), also known as Cuban oregano or Indian borage, is a semi-succulent perennial species widely used in traditional medicine for its therapeutic and nutritional properties. While its essential oils and aromatic fraction have been extensively investigated, the characterization of its non-volatile metabolites remains limited. The aim of this study was to explore the chemical composition of fresh leaves with a focus on the non-volatile fraction. **Methods:** Fresh leaves of *C. aromaticus* were cryogenically treated with liquid nitrogen, ground, and subjected to three different extraction procedures: hydroalcoholic maceration, ethyl acetate maceration, and liquid–liquid partitioning to obtain a dichloromethane organic phase and a hydroalcoholic phase. Extracts and fractions were analyzed by HPTLC and HPLC for metabolic profiling. In addition, the Bligh–Dyer method was applied to separate polar and non-polar metabolites, which were subsequently characterized using NMR spectroscopy. **Results:** Chromatographic analyses highlighted the occurrence and distribution of organic acids, polyphenols (notably flavonoids), and proteinogenic amino acids. Spectroscopic data confirmed the presence of diverse polar and non-polar metabolites, providing a more detailed chemical fingerprint of *C. aromaticus*. This integrated approach broadened the phytochemical profile of the species beyond the well-documented essential oils. **Conclusions:** The results contribute to a better understanding of the non-volatile metabolites of *C. aromaticus*, offering novel insights into its chemical diversity. These findings highlight the potential of this plant as a valuable source of bioactive compounds, supporting its future application in nutraceutical and pharmaceutical research.

## 1. Introduction

*Coleus aromaticus* (Roxb.) Benth. is a perennial, succulent herb characterized by numerous stems. The leaves are fleshy, gray-green in color, ranging from broadly ovate to ovate-deltoid, densely pubescent, with prominent veins on the abaxial surface, and emit a characteristic odor [1]. Figure 1 shows photographs of *Coleus aromaticus*.

The genus *Coleus* belongs to the family Lamiaceae and comprises about 294 species, mainly distributed in tropical and subtropical regions of Africa, Asia, and Australia, as well as in the East Indies, the Malay Archipelago, and the Philippines. These are herbaceous plants or small shrubs, often cultivated as ornamentals for their brightly colored and decorative foliage. Some species have aromatic leaves and are employed in traditional medicine or culinary practices. Recent phylogenetic studies have redefined the boundaries of the genus, distinguishing it from *Plectranthus* and *Equilabium* [2,3,4].

Several synonyms are reported for this plant, including *Coleus amboinicus* Lour. and *Plectranthus amboinicus* (Lour.) Spreng., as well as many common names such as Cuban oregano, Indian borage, and country borage, among others.

*Coleus aromaticus* is extensively employed in the culinary field, where it is consumed raw, used as a flavoring agent in various dishes, or added to aromatize beverages.

It has a long history of use in traditional medicine: in Indian Ayurvedic practice, it is regarded as one of the botanical sources of *Pashanabheda*, a term that refers to a group of plants commonly employed for their diuretic and litholytic properties, especially in the management of urinary stones [2,3,4,5].

Leaves are also widely employed for the treatment of respiratory ailments (cough, asthma, bronchitis, sore throat, congestion) and digestive disorders (indigestion, dyspepsia, dysentery, diarrhea, colitis) [3,6]. They are used against fever, headache, epilepsy, convulsions, colic, rheumatism, whooping cough, helminthiasis, and skin infections [4]. Traditional preparations include decoctions (sometimes enriched with cardamom, cloves, and honey) and juices mixed with honey or sugar, used for colds, respiratory infections, and as carminatives [3].

From a pharmacological perspective, the leaves exhibit antioxidant, antimicrobial, antimutagenic, antitumor, radioprotective, neuropharmacological, and antiepileptic activities [4,7,8,9,10,11,12].

Overall, data in the literature suggest that this species may represent an important source of biologically active compounds. Previous studies have mainly focused on the composition of the essential oil, whereas the non-volatile fraction has been scarcely investigated. In some cases, studies report only the content of broad classes of compounds, such as flavonoids, saponins, alkaloids, or tannins, by preliminary and general assays, merely indicating their presence or absence, without identifying the individual metabolites [13]. In particular, volatile constituents have been obtained by hydrodistillation or steam distillation and characterized mainly by GC and GC–MS, including SPME-based approaches; in the essential oil high concentrations of carvacrol and thymol, together with eugenol, chavicol and eucalyptol are present [4,14].

Column chromatography allowed the identification of twelve compounds from the leaves of *C. aromaticus*, belonging to the classes of flavonoids (n = 4), benzenoids (n = 3), quinones (n = 1), steroids (n = 3) and one lignan. Their structures were elucidated using spectroscopic techniques [9].

Moreover, phytochemical screening of leaves, stems, and roots showed the presence in all organs of volatiles constituents as well as non-volatiles such as sterols/triterpenes, flavonoids, carbohydrates/glycosides and catechin tannins. Roots in particular displayed the highest contents of carbohydrates, soluble sugars and proteins, followed by leaves and stems. All organs were also rich in vitamins and contained moderate levels of essential amino acids [13].

In the present work, we aimed to characterize the non-volatile metabolites, both polar and non-polar, occurring in the leaves of *C. aromaticus*. To this end, and building upon our expertise, we analyzed proteinogenic amino acids (both essential and non-essential), organic acids, and polyphenols, by combining chromatographic methods (HPTLC and HPLC) with NMR spectroscopic techniques.

From an analytical standpoint, comprehensive phytochemical investigations benefit from the integration of complementary methodologies capable of addressing the chemical diversity and concentration range of plant metabolites. In particular, a targeted metabolomic approach allows focused characterization of selected metabolite classes with known or suspected biological relevance. Chromatographic techniques such as HPLC and HPTLC remain central in this context: HPLC provides high-resolution separation and accurate quantification of individual compounds, whereas HPTLC represents a flexible and efficient tool for phytochemical fingerprinting, comparative profiling, and semi-quantitative analysis of complex non-volatile fractions, and is especially valuable in herbal and medicinal plant research. When combined with NMR spectroscopy, which offers robust structural elucidation and unequivocal metabolite identification, this integrated analytical strategy enables reliable characterization of both polar and non-polar constituents. The effectiveness of this approach for plant metabolite profiling has been previously demonstrated in our recent work and forms the methodological basis for the present investigation of the non-volatile metabolome of *C. aromaticus* leaves [15].

## 2. Materials and Methods

### 2.1. Chemicals and Reagents

Standards and solvents were purchased from Sigma (Sigma-Aldrich, Milano, Italy). All chemicals and solvents were of analytical grade.

A complete list of all chemical reagents and standards used is provided in a dedicated paragraph in the Appendix A.

### 2.2. Samples Collection and Extract Preparation

*Coleus aromaticus* was collected at the experimental garden of the Department of Environmental Botany, Sapienza University of Rome, Italy. Fresh leaves of *C. aromaticus* were weighed, immersed in liquid nitrogen, and subsequently ground in a steel mortar with pestle. Different extraction methods were applied in order to obtain a comprehensive profile of non-volatile secondary metabolites and free proteinogenic amino acids. The procedures adopted, reported in Table 1, included: hydroalcoholic maceration, maceration in AcOEt, MeOH/DCM/H_2_O partitioning, and the Bligh–Dyer method. All procedures were performed in triplicate, yielding the following sets of samples: Bligh–Dyer extracts (BDE1, BDE2, BDE3), hydroalcoholic macerates (HAM1, HAM2, HAM3), ethyl acetate macerates (EAM1, EAM2, EAM3) and, from the MeOH/DCM/H_2_O partitioning, both organic phases (OP1, OP2, OP3) and hydroalcoholic phases (HAP1, HAP2, HAP3). To obtain the macerates and partitioning mixtures, after addition of the extraction solvents, the samples were shaken on a rotary shaker (150 rpm) at 25 °C for 24 h. Subsequently, they were then filtered through filter paper, the phases resulting from partitioning were separated, and the solvents were removed using a rotary evaporator. The hydroalcoholic macerates and the hydroalcoholic fractions obtained from liquid–liquid partition were concentrated under reduced pressure and subjected to lyophilization for 24 h to achieve complete removal of residual water.

The BD extract was obtained using a different procedure and was employed exclusively for the NMR analyses. Leaves were weighed and extracted following a modified Bligh-Dyer protocol [16]. Each sample was homogenized and grounded in a mortar under liquid nitrogen. The solvents added in order were: 3 mL of cold methanol, 3 mL of cold chloroform and 1.5 mL of water to the homogenized samples. The mixture was stirred after each solvent addition. Samples were later stored at 4 °C overnight. The samples were then centrifuged for 30 min at 4 °C, with a rotation speed of 11,000 rpm. The upper hydrophilic and lower lipophilic phases were separated and dried under nitrogen flow. Dry samples were stored at −80 °C until use.

For HPTLC analysis, an aliquot of each extract was diluted with the appropriate solvent to obtain a final concentration of 5 mg/mL. Reference standards were dissolved individually in methanol at a concentration of 1 mg/mL. All solutions were stored at 4 °C until use.

For HPLC analysis, an aliquot of each extract was diluted with the appropriate solvent to obtain a final concentration of 5 mg/mL. All samples were dissolved in HPLC-grade methanol, except for the hydroalcoholic phases, which were solubilized in a MeOH (HPLC)/H_2_O (milliQ) mixture (2:1, *v*/*v*), maintaining the same final concentration (5 mg/mL). Reference standards were individually prepared in methanol at 0.05 mg/mL, except for apigenin, which was prepared at 0.125 mg/mL. All solutions were stored at 4 °C until use.

### 2.3. HPTLC Analysis

Samples and standards were applied to HPTLC Silica Gel 60 F254 20 × 10 cm plates (Merck, Darmstadt, Germany) using the automatic sampler (ATS 4, Camag, Muttenz, Switzerland). The extracts were applied by an automated ‘spray-on’ technique under a N_2_ flow at a rate of 100 nL/s, with a volume of 6 µL per application for organic acids and flavonoids, 12 µL for amino acids, and 1 µL for standards.

Plate development was performed in an Automatic Developing Chamber 2 (ADC2, Camag, Switzerland) using the selected mobile phase for both development and saturation. Different mobile phases were applied depending on the chemical class under investigation: for flavonoids and organic acids, ethyl acetate/dichloromethane/acetic acid/formic acid/water (100:25:10:10:11 *v*/*v*); for amino acids, two different systems were employed, namely 1-butanol/acetone/acetic acid/water (7:7:2:4 *v*/*v*) and 1-butanol/acetic acid/water/formic acid (28:9:8:2 *v*/*v*). In all cases, the solvent front was allowed to migrate 7 cm from the lower edge of the plate. The selection of developmental solvents (solvent type and ratios) was based on literature references and prior expertise [15,17,18].

After development, plates were dried at 120 °C for 5 min and visualized under UV 254 nm, UV 366 nm, and white light. Densitometric analysis of flavonoids and organic acids was carried out using a TLC Scanner 4 (Camag, Switzerland) at 254, 272, and 330 nm, with UV spectra recorded between 190 and 550 nm. Derivatization was performed with natural product reagent/anisaldehyde for flavonoids and organic acids, and ninhydrin for amino acids, followed by drying at 120 °C for 5 min and documentation under UV 366 nm and white light.

### 2.4. HPLC Analysis

HPLC-DAD analyses were carried out using a Nexera XR system (Shimadzu, Kyoto, Japan) equipped with a photodiode array detector (DAD, SPD-M40), an autosampler (SIL-40C XR), a quaternary pump (LC-40D XR), a degassing unit (DGU-405), a system controller (CBM-40), a thermostated column oven (CTO-40S), and a Shim-pack Velox C18 chromatographic column (150 mm × 3.0 mm, 2.7 μm).

Samples and standards were prepared as described in Section 4.2. Each final solution (3 μL injected) was analyzed by HPLC-DAD under the following conditions: mobile phase A, Milli-Q water with 0.05% H_3_PO_4_; mobile phase B, acetonitrile with 0.05% H_3_PO_4_. The flow rate was set at 0.40 mL/min, with the following gradient program: 0–5 min, 5% B; 5–35 min, 40% B; 35–36 min, 100% B; 36–51 min, 100% B; 51–52 min, 5% B; 53–67 min, 5% B. The PDA detector acquired one UV–Vis spectrum per second over the range 190–800 nm.

### 2.5. NMR Analysis

The BD hydrophilic phase was resuspended in 0.7 mL of D_2_O containing 3-(trimethylsilyl)-propionic-2,2,3,3-d_4_ acid sodium salt (TSP, 2 mM), the lipophilic phase was resuspended in 0.7 mL of CDCl_3_ containing hexamethyldisiloxane (HMS, 2 mM), as internal and chemical shift reference standards. The samples thus prepared were analyzed by ^1^H-NMR.

For the NMR experiments a JNM-ECZ 600R (JEOL Ltd., Tokyo, Japan) spectrometer operating at the proton frequency of 600 MHz and equipped with a multinuclear gradient inverse probe head, working at 298 K, was employed. For the hydrophilic phase, monodimensional ^1^H experiments employed a presaturation pulse sequence for water suppression, using a time length of 2 s, a spectral width of 9.03 kHz and 64 k data points, corresponding to an acquisition time of 5.81 s. The pulse length of 90° flip angle was set to 8.3 μs, the recycle delay was set to 5.72 s. Similar parameters were employed for the lipophilic phase, without the presaturation sequence for water suppression. Spectral assignment was carried out based on chemical shift, multiplicity, *J* couplings, ^1^H-^1^H TOCSY and ^1^H-^13^C HSQC correlations, following the protocol reported by Sciubba et al., 2020 [18] with the bidimensional experiments following spectral parameters reported by Spinelli et al., 2022 [19]. The putative assignments were corroborated by databases and literature compilations [13,20,21].

Bidimensional spectra ^1^H-^1^H TOCSY and ^1^H-^13^C HSQC of both BD-HAP and BD-lipophilic extract are reported in Appendix A.

Quantification of metabolites was achieved by comparing integrals of non-overlapping and univocally assigned resonances (bold resonances reported in Appendix A), with the integral of the respective internal standards (for TSP 9 protons in the aqueous fraction, for HMS 18 protons in the lipophilic fraction), according to the general formula:$$C_{m}=\frac{A_{m}}{A_{IS}}\times \frac{H_{IS}}{H_{m}}\times C_{IS}$$
where *C_m_* stands for metabolite concentration, *A_m_* is the area of the metabolite signal, *A_IS_* is the area of the Internal Standard (IS) signal, *H_IS_* is the number of protons generating the IS signal, *H_m_* is the number of protons generating the metabolite signal, *C_IS_* is the concentration of the IS. Metabolite concentrations were later normalized by fresh weight of the sample and expressed in mg/100 g (reported as mean ± standard deviation). Assigned spectra with quantified metabolites are reported in Appendix A. Assigned spectral region with chosen resonances for each BD-phase extract are also reported in Appendix A.

## 3. Results

This study investigated the non-volatile fraction of *C. aromaticus* leaves, with particular focus on the content of amino acids, organic acids, and phenolic compounds. The analytical strategy was based on a combination of complementary techniques, namely HPTLC, HPLC, and NMR spectroscopy. This integrated approach, already applied in previous studies, combines the sensitivity of chromatographic techniques with the detailed structural information provided by NMR, ensuring a robust and accurate profiling of bioactive compounds.

### 3.1. Amino Acids Analysis

HPTLC analysis of amino acids was performed on all samples obtained using different extraction methods (HAM, EAM, OP, and HAP), considering all 20 proteinogenic amino acids (Figure 2). Based on current knowledge, no single mobile phase is capable of distinctly separating all amino acids, as some exhibit very similar R*f* values. For this reason, two tests with different mobile phases were carried out. Nevertheless, for some amino acids the R*f* values remained comparable, as will be discussed later.

The amino acids analyzed included alanine, arginine, asparagine, aspartic acid, cysteine, glutamic acid, glutamine, glycine, histidine, isoleucine, leucine, lysine, methionine, phenylalanine, proline, serine, threonine, tryptophan, tyrosine, and valine.

With the first mobile phase (1-butanol/acetone/acetic acid/water, 7:7:2:4 *v*/*v*), arginine (R*f* 0.08), aspartic acid (R*f* 0.24), asparagine (R*f* 0.28), and glutamic acid (R*f* 0.39) were clearly identified in the hydroalcoholic samples (HAP and HAM). However, alanine and threonine (R*f* 0.43), as well as glutamine (R*f* 0.34), proline (R*f* 0.34), and serine (R*f* 0.35), showed very similar R*f* values. Using the second mobile phase (1-butanol/acetic acid/water/formic acid, 28:9:8:2 *v*/*v*), alanine (R*f* 0.39) and threonine (R*f* 0.36) could be distinguished, confirming the presence of alanine in both HAP and HAM; whereas glutamine (R*f* 0.29), proline (R*f* 0.30), and serine (R*f* 0.33) still exhibited very similar R*f* values. Despite this, the presence of both glutamine and serine in HAP and HAM is hypothesized.

As reported in Table 2, NMR analysis of the BD-HAP revealed and quantified the presence of non-aromatic amino acids alanine, asparagine, glutamine and threonine, further confirming what was identified with HPLC. Amongst them, non-proteinogenic GABA was also found and quantified.

The presence of aspartic acid was detected in the ^1^H-^1^H TOCSY spectrum but not quantified due to resonances overlapping with malic acid in the region of 2.8 ppm. Similarly, tyrosine was observed in the aromatic region thanks to the ^1^H-^1^H TOCSY correlations but was not quantified due to strong signals overlapping. The hypothesized presence of proline and serine could not be confirmed in the BD-HAP extract, probably due to lower quantities compared to NMR sensibility or general absence in the leaf matrix.

Overall, quantified amino acids in this extract account for 5% of the total sum of quantified compounds in mg/100 g of BD-HAP.

### 3.2. Organic Acids and Phenols

Unlike the amino acid analysis, the study of organic acids and flavonoids was carried out using a single mobile phase (Figure 3). This choice is supported by previous experience, which validated its effectiveness, as the spot coloration combined with double derivatization enables reliable identification of both chemical classes under identical conditions, even in cases of very similar R*f* values. In any case, coupling with HPLC ensured the accuracy of the results.

The compounds analyzed included apigenin, caffeic acid, chlorogenic acid, cinnamic acid, gallic acid, luteolin, *p*-coumaric acid, quercetin, and rutin. Analyses performed on OP, HAP, HAM, and EAM samples clearly identified apigenin, caffeic acid, rosmarinic acid, and rutin, while the presence of quercetin was less well defined. Although gallic acid exhibits the same R*f* as rosmarinic acid (0.75), its presence can be ruled out based on the spot coloration, as gallic acid would mask the rosmarinic acid spot, and the UV spectrum, which matches that of rosmarinic acid rather than gallic acid. Furthermore, HPLC analyses did not detect the presence of gallic acid. Table 3 reports the relative abundances in the different extracts together with the corresponding R*f* values.

To confirm the HPTLC findings, HPLC analyses were performed using an optimized method described in the Materials and Methods. The separation resulted in well-resolved peaks, allowing unequivocal identification. In HAP and HAM, caffeic acid and rosmarinic acid were identified, while in EAM the presence of caffeic acid, luteolin, quercetin, and apigenin was detected (Figure 3). Retention times of the reference compounds are reported in the Appendix A. HPLC analysis also revealed a peak with the same retention time as rutin (R*t* 19.4 min); however, comparison of the UV spectra between the compound in the sample and the rutin standard showed significant differences, thereby excluding its presence. Consequently, the band previously attributed to rutin in the HPTLC analysis cannot be considered as such.

To confirm the HPTLC findings, HPLC analyses were performed using an optimized method described in the Materials and Methods. The separation resulted in well-resolved peaks, allowing unequivocal identification. In HAP and HAM, caffeic acid and rosmarinic acid were identified, while in EAM the presence of caffeic acid, luteolin, quercetin, and apigenin was detected (Figure 4). Retention times of the reference compounds are reported in the Appendix A. HPLC analysis also revealed a peak with the same retention time as rutin (R*t* 19.4 min); however, comparison of the UV spectra between the compound in the sample and the rutin standard showed significant differences, thereby excluding its presence. Consequently, the band previously attributed to rutin in the HPTLC analysis cannot be considered as such.

Several organic acids were identified and quantified in the ^1^H NMR BD-HAP, such as acetic, formic, fumaric and malic acids. Phenolic acids caffeic and rosmarinic were some of the most abundant metabolites in the extract (as reported in Table 4), respectively comprising 18% and 33% of the total sum of quantified compounds in BD-HAP.

*p*-Coumaric acid could not be detected in the BD-HAP, further confirming what was previously hypothesized as its presence in leaves. Similarly, the presence of gallic acid was also ruled out in the NMR spectra on the basis of the absence of diagnostic singlets in the 7 ppm region.

Amongst flavonoids quantified in NMR spectrum, only apigenin glycosides were found, labeled as U01 and U02 due to missing correlation in the HMBC spectrum for their glucosides. Assignation was carried out based on ^1^H signals on 6.51 and 6.59 and 7.22, 6.91 and 6.88 of C-5′,3′, 6′ and 2′. Interestingly, in this spectral region, no signals associated with ABX systems were detected (except for caffeic and rosmarinic acids), excluding the presence of rutin and quercetin glycosides in BD-HAP. This is further supported by what was observed by HPTLC and HPLC-UV spectrums (Figure 2, Table 3).

Regarding the aforementioned flavonoid, none of aglycones were not detected in ^1^H BD organic phase spectrum, due to the lower sensibility of the platform compared to the other techniques.

Carbohydrates quantified in the NMR spectrum accounted mostly for fructose and raffinose (8% and 5%, of the total metabolites extracted in the BD-HAP), followed by sucrose and glucose (Table 5). Lastly other metabolites extracted in this phase are choline and unknown compounds U03 and U04, putatively assigned as 2-hydroxy-3-methylbutyric acid and octenoic acid.

The metabolites and groups present in the BD-lipophilic phase are also reported in Table 5. The composition of this extract mainly accounts for saturated fatty acids (67% of the total sum), while the saturated moiety composes 19% of the total, divided in omega 9 (15%), omega 3 (3%) and omega 6 (0.5%). Other metabolites identified were sterols, such as β-sitosterol and camposterol and the lignan U05, putatively assigned as marlignan R from literature NMR data [22].

## 4. Discussion

### 4.1. Amino Acids Analysis

Given their chemical nature, amino acids are expected to be present in the hydroalcoholic samples (HAP and HAM) rather than in the ethyl acetate (EAM) or organic phase (OP). As shown by the chromatogram intensities of HAP and HAM, analyzed at the same concentration, HAP displays a higher amino acid content. This can be attributed to the liquid–liquid partitioning step used to obtain HAP: during this process, less polar constituents are selectively transferred into the OP fraction, enriching the remaining hydroalcoholic phase in polar compounds. In addition, the exposure to the organic solvent can disrupt cellular and subcellular structures more effectively than simple maceration, thereby promoting the release of polar metabolites normally retained within cells or organelles. Consequently, HAM, being a crude hydroalcoholic macerate, contains a broader and less selective range of metabolites, including semipolar and moderately apolar compounds, resulting in a lower relative abundance of amino acids compared with the polarity-enriched HAP fraction.

El-Hawary et al. (2012) identified glycine, histidine, isoleucine, leucine, lysine, phenylalanine, tyrosine, and valine, among others, in leaves, stems, and roots [13]. In that study, however, amino acids were determined using a spectrophotometric method after acid hydrolysis; therefore, they did not represent free amino acids, as in our case, but rather originated from the degradation of proteins and enzymes present in plant tissues. Overall, the 2012 study reported the presence of 15 out of the 20 proteinogenic amino acids, and currently represents the only available work in which the amino acid composition of this species has been investigated [23]. In comparison with that study, we also identified asparagine and glutamine, as also supported by NMR analysis described below.

In addition to proteinogenic amino acids, the presence and quantification of the non-proteinogenic amino acid GABA is particularly noteworthy. GABA is widely distributed in plant tissues, where it is mainly synthesized via the decarboxylation of glutamate through the GABA shunt [24,25]. GABA is thought to play a role in pathways involved in responses to abiotic and biotic stress, contributing to cytosolic pH regulation and redox balance, as well as promoting cellular protection by enhancing antioxidant systems and maintaining membrane integrity and metabolic stability [25,26,27]. It is also considered a metabolic signal involved in cell-to-cell communication and adaptation to abiotic stimuli [26]. Emerging evidence further suggests that GABA participates in modulating ion fluxes and membrane depolarization, thereby influencing growth responses under environmental constraints [28]. Overall, GABA contributes to the regulation of key processes such as growth, root development, and stomatal opening, representing an important physiological indicator of the plant’s status [26].

### 4.2. Organic Acids and Phenols

The results obtained are consistent with the scarce data available in the literature for this botanical species, except for *p*-coumaric acid and rutin [5,29]. In the first case, it can be hypothesized that in other studies the plant material analyzed included stems or a mixture of leaves and stems, and not only leaves as in our case. *p*-Coumaric acid is a key hydroxycinnamic acid in the phenylpropanoid pathway: a precursor of lignin, flavonoids, and other structural phenolics [30]. It also takes part in defense and stress responses, modulating ROS and hormonal signaling [31,32], but may also exert allelopathic effects [33]. Given these premises, particularly its role as a lignin precursor, it is plausible as we hypothesized that it is not present in the leaves but rather in the stems.

In the second case, as highlighted by both chromatographic techniques, a spot with the same R*f* and coloration as rutin was observed in HPTLC, while in HPLC a peak with the same retention time but a different UV spectrum was detected. Analysis of the UV–Vis spectra helps clarify this discrepancy: apigenin shows two absorption maxima at 267 and 337 nm, rutin displays three characteristic peaks at 204, 257 and 354 nm, and quercetin presents maxima at 202, 225 and 371 nm (The UV spectra are available in the Appendix A). The unknown compound, despite sharing the same R*t* as rutin, exhibits a UV spectrum with only two peaks at 262 and 348 nm, indicating that it does not match either rutin or quercetin. We therefore consider it plausible that the compound in question is a structurally related glycoside based on R*t* similarity.

Moreover, considering that rutin consists of the flavonol quercetin (aglycone) linked to the disaccharide rutinose at position 3, and that its spectrum shows shifts compared to that of the aglycone quercetin, very similar to those observed between the unknown compound and apigenin, we hypothesize for the compound in question to be an apigenin glycoside. The spectral shifts observed in the UV spectrum are indeed consistent with the introduction of a sugar unit onto the flavonoid backbone.

In light of the NMR-based organic acids and phenols profile obtained in this study, overall discrepancies between the NMR-derived metabolite profile and literature data [34] may be attributed to genetic or pedoclimatic differences among plants grown under different conditions, to differences in analytical focus (e.g., volatile versus non-volatile compounds), or to metabolite concentrations below the detection limits of the analytical platform employed here.

## 5. Conclusions

Previous studies on *C. aromaticus* have not provided a comprehensive characterization of its non-volatile constituents. This gap is likely due to the plant’s strong and distinctive aroma, typical of the Lamiaceae family, which has primarily directed research toward its volatile fraction. However, owing to its aromatic properties, *C. aromaticus* is widely used in culinary applications, where the entire leaf is consumed, and it is also well known for its use in traditional medicine for the treatment of various ailments. Its long history of consumption and favorable safety profile make it an interesting candidate as a source of bioactive compounds with pharmaceutical and nutraceutical potential.

The present work focused on the non-volatile metabolite profile of *C. aromaticus*, with particular attention to free amino acids, organic acids, and flavonoids.

The analysis of free amino acids revealed the presence of eight proteinogenic amino acids, including two essential ones. Moreover, NMR analysis confirmed the occurrence of GABA—a metabolite of notable neuroactive relevance in humans.

Regarding organic acids and flavonoids, the study revealed a remarkably high content of rosmarinic acid, which represents the major phenolic component of the plant. Rosmarinic acid, a phenolic compound biosynthesized from tyrosine and phenylalanine, is widely distributed within the Lamiaceae and is well known for its antioxidant, anti-inflammatory, neuroprotective, and hypolipidemic activities. Several studies, both preclinical and clinical, have confirmed its beneficial effects on inflammation, metabolism, and cognitive functions [35]. Furthermore, rosmarinic acid has been shown to inhibit cancer cell proliferation, induce apoptosis, and prevent metastasis through multiple mechanisms, many of which are related to its antioxidant activity.

In addition to rosmarinic acid, other bioactive phenolic compounds such as caffeic acid, apigenin, luteolin, and quercetin were identified, all known for their antioxidant, anti-inflammatory, and cytoprotective properties. However, the predominance of rosmarinic acid in *C. aromaticus* indicates that it could contribute significantly to the overall biological potential of the plant.

Overall, this study highlights the added value of combining targeted chromatographic techniques (HPLC and HPTLC) with an untargeted NMR-based metabolomic approach, indeed enhancing the robustness of metabolite characterization. Taken together, the results indicate that *Coleus aromaticus* represents a promising natural source of bioactive compounds of pharmaceutical and nutraceutical interest, with rosmarinic acid emerging as the key metabolite of particular relevance.

## Figures and Tables

**Figure 1 metabolites-16-00015-f001:**
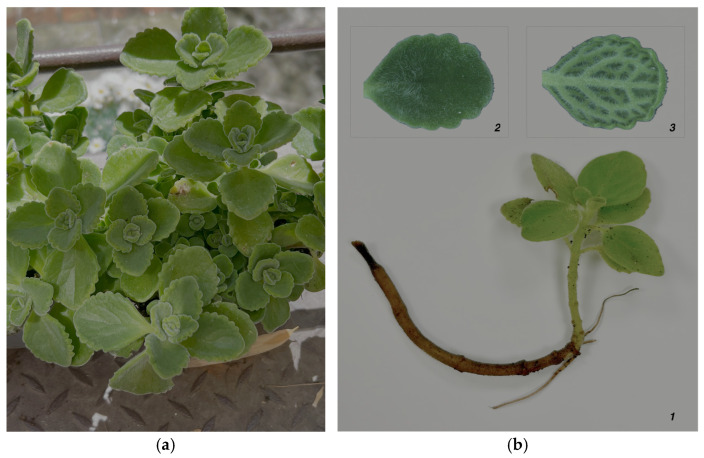
(**a**) Photograph of the Cuban oregano plant (*Coleus aromaticus*); (**b**) Image of a stem with leaves of *C. aromaticus* (1), with details of the adaxial (2) and abaxial (3) leaf surfaces. The leaf details were captured by Dr. Duilio Iamonico using a Leica M205 C stereomicroscope (Leica Microsystems Srl, Milano, Italy) at 0.78× magnification.

**Figure 2 metabolites-16-00015-f002:**
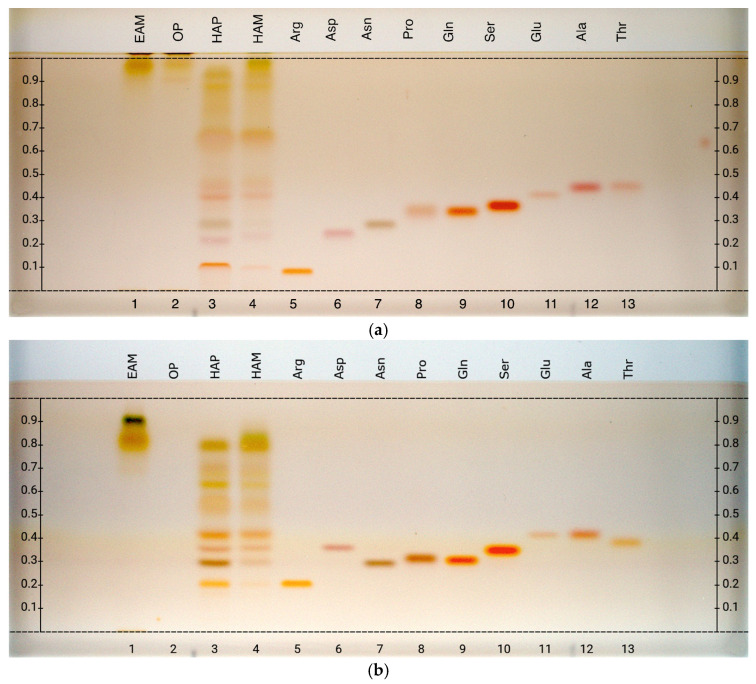
HPTLC fingerprints of amino acids from hydroalcoholic phase of *C. aromaticus* samples, visualized under white light after derivatization with ninhydrin reagents. Mobile phase: (**a**) 1-butanol/acetone/acetic acid/water (7:7:2:4 *v*/*v*), (**b**) 1-butanol/acetic acid/water/formic acid (28:9:8:2 *v*/*v*). Track assignments: 1 ethyl acetate macerates (EAM), 2 organic phase (OP), 3 hydroalcoholic phase (HAP), 4 hydroalcoholic macerate (HAM), 5 arginine, 6 aspartic acid, 7 asparagine, 8 proline, 9 glutamine, 10 serine, 11 glutamic acid, 12 alanine, 13 threonine. For clearer interpretation of the HPTLC plate, the y-axis shows the R*f* values and the x-axis the track numbers.

**Figure 3 metabolites-16-00015-f003:**
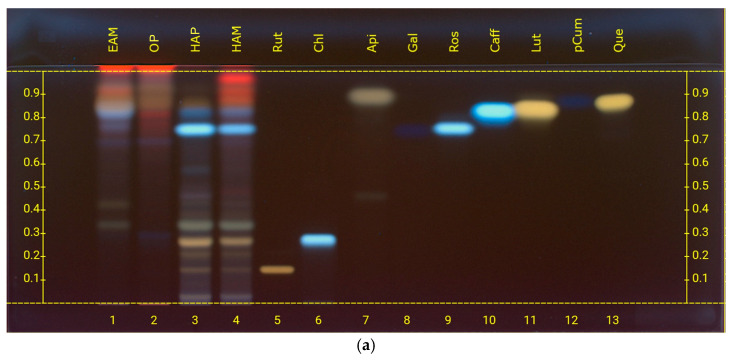
HPTLC fingerprints of flavonoids and organic acids from *C. aromaticus* samples, visualized under UV 366 nm (**a**) after derivatization with natural product reagent (**b**) after derivatization with anisaldehyde. Mobile phase: ethyl acetate/dichloromethane/acetic acid/formic acid/water (100:25:10:10:11, *v*/*v*). Track assignments: 1 ethyl acetate macerates (EAM), 2 organic phase (OP), 3 hydroalcoholic phase (HAP), 4 hydroalcoholic macerate (HAM), 5 rutin (Rut), 6 chlorogenic acid (Chl), 7 apigenin (Api), 8 gallic acid (Gal), 9 rosmarinic acid (Ros), 10 caffeic acid (Caff), 11 luteolin (Lut), 12 *p*-cumaric acid (pCum), 13 quercetin (Que). For clearer interpretation of the HPTLC plate, the y-axis shows the R*f* values and the x-axis the track numbers.

**Figure 4 metabolites-16-00015-f004:**
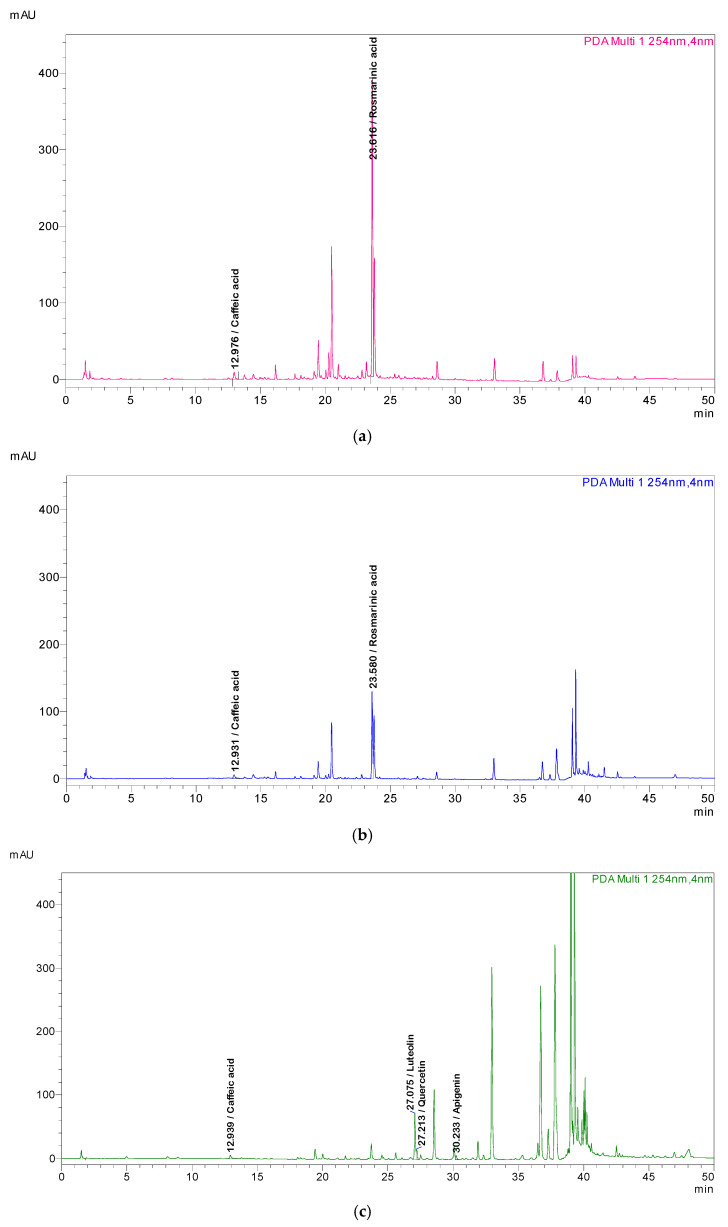
HPLC chromatograms of the different *C. aromaticus* extracts: (**a**) hydroalcoholic phase (HAP); (**b**) hydroalcoholic macerate (HAM); (**c**) ethyl acetate macerate (EAM). Identified compounds are highlighted according to their retention times (min): caffeic acid 12.97 (254 nm); rosmarinic acid 23.61 (254 nm); luteolin 27.07 (254 nm); quercetin 27.21 (254 nm); and apigenin 30.23 (254 nm).

**Table 1 metabolites-16-00015-t001:** Extraction protocols applied to *C. aromaticus* leaves.

Method	Fresh Leaves	Solvents	Agitation	T/t
Bligh–Dyer (BD)	1.2 g	3 mL MeOH + 3 mL CHCl_3_ (vortex), then 1.5 mL water	Vortex 30 s; static overnight	4 °C/12 h
Hydroalcoholic maceration (HAM)	6 g	MeOH:H_2_O = 7:3 30 mL	Rotary shaker, 150 rpm	25 °C/24 h
Ethyl acetate Maceration (EAM)	6 g	Ethyl acetate 30 mL	Rotary shaker, 150 rpm	25 °C/24 h
MeOH/DCM/H_2_O partitioning (HAP + OP)	6 g	30 mL MeOH, 30 mL DCM, 15 mL H_2_O	Rotary shaker, 150 rpm	25 °C/24 h

**Table 2 metabolites-16-00015-t002:** Quantification of the amino acids from the ^1^H NMR spectrum of hydroalcoholic extract of *C. aromaticus* leaves.

Compound	Amount (mg/100 g)
Amino Acids	
Alanine	0.56 ± 0.08
Asparagine	7.51 ± 4.65
γ-Aminobutyric acid (GABA)	1.73 ± 0.77
Glutamic acid	2.92 ± 0.58
Glutamine	0.46 ± 0.19
Threonine	0.16 ± 0.05

**Table 3 metabolites-16-00015-t003:** The flavonoid and organic acid content identified by HPTLC from *C. aromaticus* samples. Quantitative data indicated with a “+” sign are based on the comparison of the relative concentrations of the same metabolite across different samples, as inferred from signal intensity. A “-” indicates absence.

Flavonoid	EAM	OP	HAP	HAM	R*f*
Apigenin	+	+	-	-	0.89
Luteolin	-	+	-	-	0.83
Quercetin	+	+	-	-	0.86
Rutin	-	-	X *	X *	0.15
**Organic Acid**	**EAM**	**OP**	**HAP**	**HAM**	**R*f***
Caffeic acid	+++	-	+	+	0.83
Chlorogenic acid	-	-	-	-	0.27
Gallic acid	-	-	-	-	0.75
*p*-cumaric acid	-	-	-	-	0.86
Rosmarinic acid	-	-	++	+	0.75

* Rutin appears to be present since a spot with the same color and R*f* was observed; however, the UV spectrum of the spot differed from that of the reference standard. “-” indicates absence, while “X” indicates presence that still needs to be verified.

**Table 4 metabolites-16-00015-t004:** Quantification of the organic acids and flavonoids from the ^1^H NMR spectrum of hydroalcoholic extract of *C. aromaticus* leaves.

Compound	Amount (mg/100 g)
**Organic Acids**	
Acetic Acid	0.19 ± 0.03
Caffeic Acid	49.66 ± 2.65
Formic Acid	0.28 ± 0.11
Fumaric Acid	0.61 ± 0.05
Malic Acid	68 ± 19.5
Rosmarinic Acid	88.89 ± 14.09
**Flavonoids**	
U01 (Apigenin glycoside 1)	6.16 ± 0.51
U02 (Apigenin glycoside 2)	5.49 ± 0.33

**Table 5 metabolites-16-00015-t005:** Quantification of carbohydrates and polyols, lipids and sterols, other metabolites and unknown compounds, from the ^1^H NMR spectrum of hydroalcoholic and lipophilic extracts of *C. aromaticus* leaves. U03, U04 and U05 are putative assignations of molecules whose full HSQC and HMBC correlation could not be detected.

Compound	Amount (mg/100 g)
**Carbohydrates**	
Fructose	22.56 ± 4.71
Glucose	2.62 ± 0.19
Raffinose	14.73 ± 0.93
Sucrose	6.79 ± 1.03
**Lipids and Sterols**	
β-Sitosterol	0.97 ± 0.51
Campsterol	1.57 ± 0.25
Monounsaturated ω-9 fatty acid (ω-9 FA)	33.03 ± 5.71
Polyunsaturated ω-6 fatty acid (ω-6 FA)	0.97 ± 0.22
Polyunsaturated ω-3 fatty acid (ω-3 FA)	6.87 ± 1.68
Saturated fatty acid (SFA)	141.11 ± 17.54
**Other Metabolites**	
Choline	1.79 ± 0.37
**Unknown Compounds**	
U03 (2-Hydroxy-3-methylbutyric acid)	2.73 ± 0.97
U04 (Octenoic acid)	1.56 ± 0.25
U05 (Marlignan R)	27.08 ± 4.59

## Data Availability

No additional data are available in public repositories or on other platforms.

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
