# Peer review of "Multi-Analytical Insight into the Non-Volatile Phytochemical Composition of Coleus aromaticus (Roxb) Benth"

_metabolites, 2025, doi:10.3390/metabo16010015_

Round 1
Reviewer 1 Report
Comments and Suggestions for Authors
This manuscript (metabolites-4041555) deals with the use of multiple analytical techniques (HPTLC, HPLC DAD and NMR) to characterise the non-volatile phytochemical components in Coleus aromaticus leaves. Topic is interesting and the results are looking promising. I would recommend its publication after following issues are addressed.
Major comments
1- A part regarding the analytical methodologies used for the analysis of Coleus aromaticus (Roxb.) Benth in introduction section.
2- The authors used trap column before analytical column ?
3- The number of replications is given, but statistical variance (e.g. standard deviation or percentage relative standard deviation) is only reported in limited detail.
4- The method is sound but data should be backed up by statistics.
5- NMR correlations should be presented more clearly.
6- It would be better to associate the study with biological activities, which are more relevant to the subject.
Minor comments
1- The supplementary reference order should match the main text references.
2- Please standardise the figure axes and unit labels.
Author Response
Major comments
1- A part regarding the analytical methodologies used for the analysis of Coleus aromaticus (Roxb.) Benth in introduction section.
Thank you for the comment. The Introduction section has been revised and implemented as requested; the newly added and revised text is highlighted in red in the manuscript.
2- The authors used trap column before analytical column?
For the HPLC analyses, trap columns were not used. Prior to analysis, the samples were filtered using a syringe filter equipped with a hydrophilic PTFE membrane (0.22 μm). As also reported in the text at line …
3- The number of replications is given, but statistical variance (e.g. standard deviation or percentage relative standard deviation) is only reported in limited detail.
Thank you for the suggestion, we amended the paragraph material and methods specifying how we evaluated statistical variance.
4- The method is sound but data should be backed up by statistics.
Regarding the NMR technique, a single extraction method (Bligh–Dyer, BD) was applied, and no comparisons among different plant species were performed. Therefore, since no categories were available for comparison, no statistical analysis was carried out. The three technical replicates were performed solely to calculate the mean and standard deviation.
With respect to HPTLC analyses, a semi-quantitative approach was adopted based on the comparison of spot intensities of samples prepared using different solvents and applied to the plates at the same concentration. This approach allowed the identification of the solvent most suitable for extracting specific classes of compounds. For this reason, no statistical analysis was performed.
5- NMR correlations should be presented more clearly.
We would like to thank the reviewer for the suggestion. To improve readability and clarity, we added supplementary figures S5-S11 of assigned metabolites resonances for each spectral region. In this way is clearly visible the signal chosen for assignation, and in reference to supplementary table 1 we report for each metabolite all correlations and signals observed.
6- It would be better to associate the study with biological activities, which are more relevant to the subject.
Thank you for the suggestion. A subsequent phase of the project is indeed planned to include in vitro biological activity assays, such as antioxidant activity tests, total polyphenol content, and potential enzymatic activities. These evaluations will allow us to correlate the chemical profiles of the extracts with their biological relevance, thereby strengthening the overall significance of the study.
In addition, as suggested in Major Comment No. 1, we have integrated and discussed the biological activities reported in the literature, together with the corresponding analytical and bioassay methodologies.
Minor comments
The supplementary reference order should match the main text references
The table and figures in the supplementary material are now ordered by their citation order in the main text.
Please standardise the figure axes and unit labels.
Chromatograms in figure 4 now have uniformly scaled x and y axes.
Reviewer 2 Report
Comments and Suggestions for Authors
The manuscript entitled “A Multi-Analytical Insight into the Non-Volatile Phytochemical Composition of Coleus aromaticus (Roxb.) Benth” submitted to Metabolites (Manuscript ID: metabolites-4041555) offers a deep characterization of the non-volatile metabolites present in the fresh leaves of Coleus aromaticus, with particular attention to free amino acids, organic acids, and flavonoids.
After immersion of the fresh leaves of C. aromaticus in liquid nitrogen, and subsequently grounding, the samples were subjected to different extraction protocols: hydroalcoholic maceration, maceration in ethyl acetate, MeOH/DCM/Hâ‚‚O partitioning, and the Bligh–Dyer method. The extracts and fractions were analyzed by combining chromatographic methods (HPTLC and HPLC) with NMR spectroscopic techniques: HPTLC and HPLC for amino acids analysis; HPTLC and HPLC for the study of organic acids and flavonoids.
This study highlights the potential of Coleus aromaticus as a valuable source of bioactive compounds of pharmaceutical and nutraceutical interest. The importance of combining targeted chromatographic techniques (HPLC and HPTLC) with an untargeted metabolomic approach based on NMR, for a detailed chemical fingerprint / metabolites characterization, is also reconfirmed.
The writing is pretty clear, and this manuscript should provide interesting information for readers.
I have few comments and suggestions to further improve the manuscript (please see the manuscript attached), mainly concerning minor editing/formatting and the references that should be edited according to the journal’s requirements. In the manuscript, most of the paragraphs that require attention have been indicated (by underlining them in an orange-pink color) and the suggestions/observations were inserted as sticky notes.
Also, in Supplementary materials please correct:
- for Supplementary Table 1, please revise the title given the fact that the table also contains the 13C chemical shifts;
- in Supplementary Table 1, in the last column for 13C NMR please correct “d” to “δ” ppm;
- in the legend of Supplementary Figure 2a, please precise, if possible, what is the unknown compound (U01, U02, ...???);
- number also Supplementary Figure X (it will be Supplementary Figure 5), and renumber the following figures;
- starting with Supplementary Figure 3, rearrange the titles under the figures (as you already done for Supplementary Figures 1 and 2).

Author Response
We thank the Reviewer for the careful evaluation of the manuscript and for the constructive comments provided. All the suggested revisions and the reported editing and formatting issues have been carefully addressed and corrected throughout the manuscript and the Supplementary Materials, in full compliance with the journal’s guidelines.
Regarding apigenin, this standard was prepared at a different concentration due to solubility constraints.
The identification and quantification of amino acids by HPLC were not performed because the DAD detector does not allow effective UV detection of aliphatic amino acids, which lack suitable chromophores. For this reason, HPTLC, despite the intrinsic limitation related to partial spot overlapping, allows a more comprehensive visualization of the amino acid profile, especially when considering both UV-inactive and UV-active compounds.
Reviewer 3 Report
Comments and Suggestions for Authors
The authors have done a commendable, high-quality work on the multi-analytical phytochemical composition of C. aromaticus. This manuscript's multi-analytical approach is scientific, and the methods utilized to evaluate the non-volatile chemical constituents is well recommended. The manuscript is ready for publication in Metabolites post-addressing these MINOR comments:
- To enhance readership, it might be wise to also add an image or two of the C. aromaticus
- Reference no. 3 does not match content uniformity
- Recheck the entire manuscript for any font size and grammatical errors
Author Response
1- To enhance readership, it might be wise to also add an image or two of the C. aromaticus
Thank you for the suggestion. Images of C. aromaticus have now been included, comprising photographs of the whole plant as well as stereomicroscope images. In addition, a graphical abstract has also been prepared and added to further enhance the visual appeal and readability of the manuscript.
2- Reference no. 3 does not match content uniformity
It has been formatted.
Round 2
Reviewer 1 Report
Comments and Suggestions for Authors
The authors addressed all my concerns in the revised manuscript and I would recommend its publication in the journal.